# Cloning and Expression of the Tibetan Pig Interleukin-23 Gene and Its Promotion of Immunity of Pigs to PCV2 Vaccine

**DOI:** 10.3390/vaccines8020250

**Published:** 2020-05-26

**Authors:** Yongle Xiao, Huan Zhang, Jianlin Chen, Yi Chen, Jinghai Li, Tingyu Song, Guangzhi Zeng, Xiaohui Chen, Xuebin Lü, Pengfei Fang, Rong Gao

**Affiliations:** 1College of Life Sciences, Sichuan University, Chengdu 610065, China; xiao_yongle@163.com (Y.X.); chenyizoe@163.com (Y.C.); stytiki@163.com (T.S.); 2School of Basic Medical Sciences, North Sichuan Medical College, Nanchong 637000, China; zhanghuan@163.com; 3School of Laboratory Medicine, Chengdu Medical College, Chengdu 610500, China; jianlin1126@163.com; 4Sichuan Huapai Biopharmaceutical Company, Chengdu 610026, China; jinghaili@126.com (J.L.); hanlinxyc@163.com (G.Z.); pengffeif@163.com (P.F.); 5Sichuan Animal Science Academy, Chengdu 610066, China; xiaohchen@yeah.net (X.C.); lxuebin@yeah.net (X.L.)

**Keywords:** pig, interleukin-23 gene, chitosan nanoparticles, PCV2 vaccine, immune response

## Abstract

Vaccines against Porcine circovirus type 2 (PCV2) have been studied intensely and found to be effective in decreasing mortality and improving growth in swine populations. In this study, interleukin-23 (IL-23) gene was cloned from peripheral blood mononuclear cells (PBMCs) of Tibetan pigs and inserted into a eukaryotic VR1020 expression vector-VRIL23. Coated with chitosan (CS), the VRIL23-CS was intramuscularly injected into 3-week-old piglets with PCV2 vaccine. The blood was collected after vaccination at 0, 1, 2, 4, 8, and 12 weeks, respectively, to detect the immunological changes. The IgG2a and specific PCV2 antibodies were detected using ELISA, and blood CD4+ and CD8+ T cells were quantified by flow cytometry. Quantitative fluorescence PCR was used to evaluate the expression of immune genes. The results indicate that leukocytes, erythrocytes, and CD4+ and CD8+ T cells increased significantly in the blood of VRIL23-CS inoculated piglets in comparison with the control (*p* < 0.05) and so did the IgG2a and PCV2 antibodies. In addition, the expressions of Toll-like receptor (TLR) 2, TLR7, cluster of differentiation (CD) 45, IL-15, IL-12, signal transducer and activator of transcription (STAT)1, STAT2, STAT3, STAT4, and B-cell lymphoma (Bcl)-2 genes were also obviously higher in the VRIL23-CS inoculated pigs at different time points (*p* < 0.05). Overall, the results demonstrated that VRIL23-CS can enhance the comprehensive immune responses to PCV2 vaccine in vivo and has the promising potential to be developed into a safe and effective adjuvant to promote the immunity of pig against PCV disease.

## 1. Introduction

Porcine circovirus type 2 (PCV2) causes post-weaning multisystemic wasting syndrome (PMWS), which results in a major economic loss for the pork industry worldwide [1]. PCV2 infections are frequently reported in many pig farms in China [2]. Due to intensive pig production, mixed infections with other pathogens, such as porcine reproductive and respiratory syndrome virus (PRRSV), classical swine fever virus (CSF), porcine parvovirus, and mycoplasma, further worsen the situation for animal disease control [3,4]. So far, the most effective measure to control PCV2 infection has depended mainly on vaccination; however, the immunologic potency of available vaccines is still inadequate, and development of effective adjuvant remains a priority. Indeed, a few reagents, including IFN-α [5], IL-2, IL-4, IL-6 [6], IL-12 [7], α-galactosylceramide (α-GalCer) [8], CpG motifs [9], and chitosan oligosaccharides (COS) [10], have been reported to serve as vaccine adjuvants of porcine diseases.

Interleukin-23 (IL-23) is a heterodimeric cytokine composed of a p40 subunit that is shared with IL-12 and a unique p19 subunit. In the active form of IL-23, the p19 subunit is covalently linked to p40 [11], and it is in this form that it engages with its receptor (IL-23R), which is present on both innate and adaptive immune cells, including macrophages, monocytes, dendritic cells, type 3 innate lymphoid cells (ILC3s) [12], Th17 and Th22 cells [13], and neutrophils [14]. IL-23 is best known for its ability to promote Th17 maturation [15]. Specifically, IL-23 expands Th17 cells and contributes to their cytokine production, including IL-17A, which is the major effector molecule of Th17 cells [16]. Modulating Th17 responses is thought be a mode for adjuvant development [17]. Despite the importance of IL-23, to date, there are few studies to investigate the effect of IL-23 on antiviral immunity in pig disease.

The Tibetan pig is a unique native breed of the Western Qinghai-Tibetan Plateau of China and has evolved to adapt to a roughage diet and harsh environment while manifesting a strong disease resistance [18]. Here, we report the cloning of the Tibetan pig IL-23 gene and its expression with recombinant plasmid. We further analyze its effect on the in vivo immune responses of piglets to PCV2 vaccine, thereby presenting evidence for its potential value in facilitating the prevention of PCV2 disease.

## 2. Materials and Methods

### 2.1. Cloning of Tibetan Pig IL-23 Gene 

#### 2.1.1. Cloning of p40 and p19 Subunit Genes from Pig Peripheral Blood Mononuclear Cells (PBMCs)

The PBMCs from Tibetan pigs (provided by Sichuan Academy of Animal Science, Chengdu, China) were isolated via lymphocyte separation kit (Tian Jin Hao Yang Biological Manufacture, Tianjin, China) and cultured with 1 μg/mL of lipopolysaccharide (LPS) (Sigma Chemical Co., St. Louis, MO, USA) for 4 h at 37 °C. RNAiso Plus (TaKaRa Bio Inc., Kyoto, Japan) was then used for total RNA extraction and converted to cDNA using the TaKaRa PrimeScript RT reagent kit (TaKaRa Bio Inc., Kyoto, Japan). To amplify the target gene, primers for PCR were designed from the 5′ and 3′ regions of porcine p40 (GenBank accession number U08317) and p19 (GenBank accession number AB521204) subunit genes. The specific primer sequences are listed in Table 1. The resulting PCR products were subjected to separation by agarose gel electrophoresis and cloned into the pEASY-T1 vector (Transgene Biotech, Beijing, China). The clones of both the p40 and p19 subunits were completely sequenced.

#### 2.1.2. Construction of the Recombinant Eukaryotic Expression Plasmid

Linkage of p40 and p19 was achieved by use of the 63bp 2A self-cleaving sequence (GGA AGC GGA GAG GGC AGG GGA AGT CTT CTA ACA TGC GGG GAC GTG GAG GAA AAT CCC GGG CCA), with a downstream tissue plasminogen activator (TPA) signal sequence to produce a biologically active IL-23. The fused fragment was subcloned into the eukaryotic secretory expression plasmid VR1020 (Vical company, San Diego, CA, USA) using the In-Fusion cloning technique (TaKaRa Bio Inc., Kyoto, Japan). PCR, enzyme digestion, and sequencing were used to validate the IL-23 recombinant plasmid (VRIL23).

### 2.2. Expression of VRIL23 and Bioactivity In Vitro

HEK293 cells were cultured in 6-well plastic plates, followed by transfection with plasmid VRIL23 (2 μg/well). Green fluorescent protein (GFP) plasmid was used as positive control and VR1020 and phosphate buffer saline (PBS) as negative control. The transfected cells were harvested at 24, 48, and 72 h respectively. RNA was extracted from the 48 h cell, and reverse transcription-polymerase chain reaction (RT-PCR) was used to assess mRNA levels. The cell supernatant of three time points was used to stimulate the lymphocyte proliferation. The detailed steps were as follows: the PBMCs from Tibetan pigs was isolated as the above and cultured with 5 μg/mL Con A (Sigma Chemical Co., St. Louis, MO, USA) in 1640 medium for 24 h, and the blast cells were harvested and washed three times by centrifuge and then cultured in 1640 medium containing 20 mg/mL Methyl-α-D-mannoside (Sigma Chemical Co., St. Louis, MO, USA) in 6 × 10^6^/mL. The treated PBMCs were added to 96-well plate (50 μL/well) along with equivalent supernatant from transfected human embryonic kidney 293 (HEK293) cell. Each sample was divided into three duplicate wells. The culture vessels were incubated at 37 °C in an atmosphere of 5% CO_2_ for 48 h. Then 10 mL 2-(2-methoxy-4-nitrophenyl)-3-(4-nitrophenyl)-5-(2,4-disulfophenyl)-2H-tetrazolium sodium salt (WST)-8 in cell counting kit (CCK)-8 was added to each well, and the cells were cultured under the same condition for another 2 h. OD450 was determined with Microtiterplate Reader 680 (Bio-Rad, Hercules, CA, USA).

### 2.3. Effect of VRIL23 Chitosan Nanoparticles on PCV2 Vaccination In Vivo

#### 2.3.1. Preparation of Chitosan Nanoparticle-Encapsulated VRIL23

Chitosan (CS; 95% deacylated, Mw: 150kDa) was purchased from the Chengdu Organic Chemistry Institute of China Academy of Science (Chengdu, China). The VRIL23 encapsulated CS was generated via ionotropic gelation. Briefly, CS (2.4 mg/mL in 1% acetic acid, pH 5.5) was passed through a 0.22 μm filter, combined with sodium polyphosphate and incubated for 20 min at 55 °C. The plasmid solution was gradually added, dropwise, to the CS solution with stirring in a water bath at 55 °C until the mass ratio (CS:plasmid) was 30:1. The formed particles were then maintained at 55 °C for 10 min. The zeta electricity potential, particle size, and dispersion degree of the CS-entrapped VRIL23 (VRIL23-CS) were measured using a Hydro 2000MU wet sample dispersion unit (Malvern, UK). Transmission electron microscopy was used to visualize the particles. The endotoxin level of VRIL-23 CS was analyzed by the standard coagulation of Limulus Amebocyte lysate test, and the content of endogenous toxin in the preparations was found to be less than 0.1 EU/mg.

#### 2.3.2. Animal Immunization

Ten healthy piglets (hybrids of Landrace, Yorkshire, and Duroc) aged 21 days were provided by Sichuan HuaPai Biopharmaceutical (Chengdu, China). All were tested for PCV2, PCV3, mycoplasma, classical swine fever virus (CSFV), and PRRSV using both ELISA and RT-PCR and found negative. Animals were then equally randomized into two groups, with one group receiving an intramuscular injection of 2.6 mL VRIL23-CS (0.5 mg/mL, 1.5 mg recombinant plasmids prepared as above) and the other group with saline as a control. At the same time, both groups were intramuscularly injected with 2.5 mL inactivated PCV2b vaccine (Sichuan HuaPai Bio-pharmaceutical Co., Ltd., Chengdu, China) in this experiment. It contains the most epidemic virulent PCV2b virus strain in China, which was isolated and cloned in Zhejiang province, named as ZJ/C strain. One dose for pig vaccination includes the equivalent of 10^8.0^ tissue culture infective dose (TCID) 50/mL of inactivated strain of PCV2b. Both groups of piglets were raised separately under the same feeding and management conditions. Blood samples (4 mL) were then taken from the precaval vein from each animal on days 7, 14, 28, 56, and 84 post inoculation. All animals were treated in a manner consistent with Chinese animal welfare laws, guidelines, and regulations. The protocol was approved by the institutional animal ethics committees of Sichuan University (SYXK (chuan)-2018-185).

##### Peripheral Hemocytology Assay

Samples (500 μL) of peripheral blood collected in (Ethylene Diamine Tetraacetic Acid) EDTA were analyzed for blood parameters (erythrocyte, leukocyte, hemoglobin, and platelet counts) in a MIND-RAY BC-5000 hematology analyzer (Shenzhen Mindray Biomedical Electronics, Shenzhen, China).

##### CD4+ and CD8+ T Cells Assessment

Monoclonal antibodies for flow cytometry (mouse anti-porcine CD4-PE and CD8a-FITC) were produced by Southern Biotech (Birmingham, AL, USA). A total of 100 μL of porcine blood was mixed with 50 μL saline and incubated with 2 μL of anti-CD4 and 2 μL of anti-CD8a for 30 min in the dark at ambient temperature. Following incubation with 600 μL (10% *v*/*v*) lysing solution (BD, Franklin Lakes, NJ, USA) for 10 min, the remaining cells were washed twice in PBS, centrifuged at 500g for 5min, resuspended in a total volume of 300 μL PBS, and assayed in a FACScan flow cytometer (BD, USA). First electronic gate was set on lymphocytes and then gated CD4+ T cells in Y axis and CD8+ T cells in X axis as Figure 1. Dead cells were excluded according to the forward scatter and side scatter. 

##### Antibody and IgG2a

Specific antibodies against PCV2 and porcine IgG2a were measured used quantitative detection ELISA kits from Minxin Biology Co., Ltd, Chengdu, China, according to provided protocols. ODs were measured at 450 nm in a microtiterplate reader 680 (Bio-Rad, Hercules, CA, USA).

##### Immune Gene Expression

Blood (100 μL) was combined with 1 mL RNAios Plus (TaKaRa, Kyoto, Japan) for total RNA extraction using the manufacturer’s protocol. The TransScript All-in-One First-Strand cDNA Synthesis SuperMix (Transgen, Beijing, China) kit was used for synthesizing cDNA based on the provided protocol.

Primers for immune-related genes were designed according to GenBank sequence searches using Primer 5.0 and synthesized by the Tsingke Company (Beijing, China). All the primer sequences and annealing temperatures are shown in Table 2. Peptidylprolylisomerase A (PPIA) served as the normalization control. The qRT-PCR was performed on an iQ5 Applied Biosystems (Bio-Rad, Hercules, CA, USA) in a total reaction volume of 15 μL containing 1 μL cDNA, 7.5 μL SsoAdvance^TM^ Universal SYBR Green Supermix (Bio-Rad, Hercules, CA, USA), and 0.25 μL forward and reverse primers. The qRT-PCR reaction was as follows: 30 s at 95 °C, then 40 cycles of 5 s at 95 °C, and 30 s at an appropriate annealing temperature, after which a melt curve was generated, with a cycle of 65–95 °C and 0.5 °C per 5 s. Relative expression was determined by the 2^−∆∆Ct^ method.

### 2.4. Statistical Analysis

GraphPad Prism6 (Graphpad Software, Hercules, CA, USA) was used for data management and analysis. Comparisons between two groups were performed using the two-tailed Student’s *t*-test. *p* < 0.05 was set to be significant.

## 3. Results

### 3.1. Cloning of p40 and p19 Subunit Genes

cDNAs of the Tibetan pig IL-23 p19 and IL-23 p40 subunits were cloned and sequenced. The length of p19 subunit is 582bp, coding for 193 amino acids, and the p40 is 972bp for 323 amino acids (Figure 2A). The sequences have been deposited in GenBank with accession numbers KF246515 and KF246516.

### 3.2. Expression of VRIL23 and Its Bioactivity In Vitro

Figure 2B shows the RT-PCR of VRIL23 transfected cells. Both p40 and p19 subunits of IL-23 were expressed, and the length of mRNA was about 1700 bp. GFP was also successfully expressed in HEK293 cells (Figure 3). Cell viability assay by CCK8 is shown in Figure 4. It is apparent from this figure that IL-23 expressed in HEK293 cell can induce remarkable lymphocyte activation and cell proliferation (*p* < 0.05). 

### 3.3. Chitosan Nanoparticle-Encapsulated VRIL23 Detection

VRIL23 was encapsulated in chitosan nanoparticles with an average diameter of 109.6 nm, and the zeta potential of the nanoparticles was +24.5 mV (Figure 5). Figure 6 shows the spherical shape of chitosan nanoparticles under the transmission electron microscope.

### 3.4. Changes in Peripheral Blood Immune Cells

The results of peripheral hemocytology assay are presented in Figure 7. From the data, we can see that the blood erythrocytes and leukocytes in the VRIL23-CS group were significantly higher than the control on 14, 35, and 84 days post vaccination. Platelet number of VRIL23-CS group was significantly lower than the control at day 7 but reached the same level after that. No significant difference of hemoglobin was found between two groups (*p* > 0.05).

### 3.5. Changes in CD4^+^ and CD8^+^ T Cells

After vaccination, the percentage of CD4^+^ T cells of the VRIL23-CS group rose significantly in comparison with the control from days 56 to 84 (*p* < 0.05). In addition, the CD8^+^ T cells showed a similar increase at days 35 and 84 (*p* < 0.05) (Figure 8).

### 3.6. Humoral Immune Changes

Figure 9 demonstrates a significant increase of IgG2a in the sera of the VRIL23-CS group compared to the control group (*p* < 0.05). During the entire period, the specific antibody against PCV2 remained higher levels in the VRIL23-CS group in comparison with the control (*p* < 0.05).

### 3.7. Immune Gene Expression

As shown in Figure 10a, TLR2 gene increased significantly in the VRIL23-CS group from days 7 to 14 after the treatment compared to the control group (*p* < 0.05). In contrast to control, the VRIL23-CS group exhibited significantly elevated expression of TLR7 and IFN-γduring the entire period (*p* < 0.05) (Figure 10b,c). Moreover, there is a clear trend that VRIL23-CS injection resulted in a significant IL-12 up-regulation compared to the control at day 7 (*p* < 0.05) (Figure 10d).

To evaluate the effect of the treatment on immune memory genes, CD45 and IL-15 were detected by QRT-PCR. Data from Figure 11 shows that the mRNA level of CD45 and IL15 markedly increased in the VRIL23-CS group compared to the control group during 84 days (*p* < 0.05). Surprisingly, this was quite similar to the expression of TLR7.

Finally, the gene expressions of immune signal transduction molecules in the pigs were examined. STAT1 gene expression in the VRIL23-CS group was significantly elevated compared to the control group during all periods post inoculation (*p* < 0.05). The expression of the STAT2 gene showed the same result except for day 56. STAT3 had the same trend except for days 14 and 84, and its level in VRIL23-CS group reached the peak at day 7. The expression of STAT4 and Bcl-2 gene in the VRIL23-CS group were markedly higher than the control group from days 7 to 14 and on day 84 (*p* < 0.05) (Figure 12).

### 3.8. Changes of Weight

Table 3 shows that the weight gain of the VRIL23-CS group remarkably increased during the 84 day period compared to the control group (*p* < 0.05). The feed–gain ratio of VRIL23-CS was obviously better than the control. There was no death for any of the animals. This suggests that the inoculation with VRIL23-CS can boost piglet growth. 

## 4. Discussion

Infectious diseases of animal are mainly prevented and controlled by vaccination. Inactivated vaccine development usually requires the use of a suitable adjuvant. Numerous reports have shown that organic, inorganic, synthetic, and natural origin compounds have the capacity to stimulate immune responses and show potent adjuvant properties [19]. These compounds are broadly divided in two different groups, namely immunostimulants, such as saponins, Toll-like receptor (TLR) agonists, and cytokines, and delivery agents, such as emulsions, microparticles, and mineral salts [20]. With regard to animal vaccine adjuvants, cytokines are being intensively investigated due to their vital and complicated effects on immune responses [21]; for instance, IL-2, IL-12, and IFN-γ can promote Th1 cell-mediated immune responses to target intracellular pathogens. Conversely, IL-4, IL-5, IL-6, and IL-10 enhance the development of Th2 cells and antibody production, which are necessary for defense against extracellular pathogens [22]. However, the use of IL-23 has yet to be explored.

In this study, for the first time, we cloned IL-23 gene from the Tibetan pig and encapsulated it into chitosan nanoparticles as an adjuvant for PCV2 vaccination of healthy pigs. We then assessed the immunological changes induced by the expression of IL-23 gene in vivo. The cloning technique used in the study was an important aspect of the creation of the IL-23 adjuvant, because both subunits of IL-23 must be generated in the same cell to facilitate secretion of intact active IL-23. This was exemplified by the fact that when either subunit alone is expressed in 293 cells, only poorly secreted non-biologically active IL-23 was produced [23]. Furthermore, there is a disulfide bond between Cys54 of p19 residue and Cys177 of p40, stabilizing the interaction between these subunits [24]. In our experiment, we used the special 2A sequence to link p40 and p19 to produce an intact active IL-23. This rational molecular design guarantees the bioactivity of secreted IL-23, which was confirmed by our in vitro lymphocytes proliferation.

The primary receptors connecting extracellular signals to p19 and p40 production are pattern recognition receptors (PRRs). As such, TLR2, 3, 4, 5, 7, and 8, C-type lectin receptors, nucleotide bingding and oligomerization domain (NOD)-like receptors, and CD40 respond to their respective ligands. We observed an up-regulation of TLR2 and 7 in the VRIL23-CS group, implying the enhanced recognition of pathogen, which also coincided with an increased IL-12 expression. IL-12 is a heterodimeric proinflammatory cytokine which is comprised of two subunits—p35 and p40. This cytokine is the main regulator of Th1 differentiation and is essential for bridging innate and adaptive immunity [25]. For instance, infection from Mycobacterium tuberculosis is more prominent in p40-deficient mice [26], and the same is true for Cryptococcus neoformans [27] in p35-deficient mice. The polypeptide p19 possesses no biological activity on its own. However, when p19 combines with the p40 subunit of IL-12, heterodimeric biologically active IL-23 is generated [23]. Interestingly, IL-12 is not produced by p35-deficient mice, but they do make IL-23. Hence, the increase of IL-12 can mirror the probable potentiation of IL-23 regulation and cellular immunity of mice against viral infection. The link between IL-12 and IL-23 should thus be further explored later.

It is very important to understand the basic immunomodulatory functions of adjuvants to show that they just enhance immunity and avoid adverse effects. IL-15 functions to promote the proliferation and differentiation of immune memory cells, increasing the CD4+ T and CD8^+^ T cells to enhance the adaptive cellular immunity [28]. In addition, CD45 on all leukocytes is vital for T cell activation via the T cell receptor (TCR) [29]. Herein, the increase of IL-15 and CD45 in the VRIL23-CS group suggests that the up-regulation of the immune memory function would facilitate the immunoprotective efficiency of PCV2 vaccine in infected piglets, which is consistent with the amplification of cellular immunity resulted from the addition of CD4+T and CD8+T cells.

The Janus Kinase (JAK) -STAT pathway plays a crucial role in T cell activation, as CD4^+^ T cells can differentiate into distinct Th1, Th2, and Th17 regulatory T cell (Tregs) subsets under the influence of different cytokines [30]. The raised expression of STAT1, 2, 3, and 4 in VRIL23-CS mice indicated the stronger activation of this signaling cascade to antigenic stimulation during PCV-2 vaccination, which is consistent with the result of another study [31], although it is contradictory with the previous research in vitro [32]. Thus, further exploration of the molecular mechanism will clarify their immune modulation mechanism and effects. Moreover, the increased expression of Bcl-2 in the VRIL-23-CS group could promote the proliferation and differentiation of competent lymphocytes during immune activation due to its anti-apoptosis effect. Consequently, this change complies with the raised levels of CD4+ T, CD8+ T cells, specific antibodies, and IgG2a, which is beneficial for the enhancement of specific humoral and cellular immunity to vaccine. Therefore, based on our observation, it is obvious that the enhancement of the JAK-STAT signaling pathway could synergize the potentiation of the adaptive immunity to vaccination.

Previous reports confirmed that chitosan is a safe and biocompatible material for gene transfection and expression in vivo [33,34,35]; hence, we selected chitosan to entrap the recombinant VRIL-23 and prepare suitable nanoparticles with positive potentials to protect the plasmid from digestion in vivo and facilitate its delivery into cells to promote the expression of IL-23. In our test, the chitosan nanoparticles were separated from the PCV2 vaccine, and moreover, its inoculation dosage was also relatively low (<3 mg per capita), which is not adequate to elicit remarkable promotion of immunity as reported by others [10,36,37]. Hence, we inferred that less chitosan of nanoparticles could not influence the release of antigen and elevate the immune response of animal as before [38,39]. Besides, due to the limited availability of health qualified piglets and instead of blank VR1020-CS, we just chose the normal saline solution as a control inoculation to investigate the entire adjuvant effect of VRIL-23-CS on pig immunity to PCV2 vaccine and intended to further reveal whether it could result in covert adverse reaction in piglet. Fortunately, no obvious injection local and systemic lesions were observed in the treated piglets during the whole experiment period. Furthermore, VRIL-23-CS-treated piglets achieved better growth performance and weight gain after vaccination compared to the control. These approved the importance of VRIL-23-CS as a safe and novel adjuvant for PCV2 vaccine.

Here, we first cloned the IL-23 gene from Tibetan pig and constructed its eukaryotic expression plasmid. The recombinant plasmid was then packed with chitosan nanoparticles and used as adjuvant for PCV2 vaccine. Many indicators consolidated that the immune function was enhanced at both innate and adaptive level, and the correlated immune memory response was also elevated. In general, therefore, it seems that the VRIL23-CS nanoparticle is a promising safe and strong adjuvant for PCV2 vaccine to promote the control of PCV infections in pig.

## 5. Conclusions

In brief, the IL-23 gene was first cloned from Tibetan pig, and its eukaryotic recombinant plasmid was constructed and expressed in vitro. The recombinant plasmid was then packed with chitosan nanoparticles and used as an adjuvant for PCV2 vaccination in piglets. The results were that IgG2a and PCV2 antibodies, leukocytes, erythrocytes, and CD4+ and CD8+ T cells increased significantly in the blood of VRIL23-CS-inoculated piglets. Besides, the expressions of TLR2, TLR7, CD45, IL-15, IL-12, IFN-γ, STAT1-4, and Bcl-2 genes were remarkably elevated in the VRIL23-CS-inoculated pigs. These results suggested that VRIL23-CS can obviously enhance the innate, humoral, and cellular immunity to PCV2 vaccine in vivo and also improve immune memory response, which would facilitate the development of VRIL23-CS as promising safe and effective adjuvant to promote the immunity of pig against PCV disease.

## Figures and Tables

**Figure 1 vaccines-08-00250-f001:**
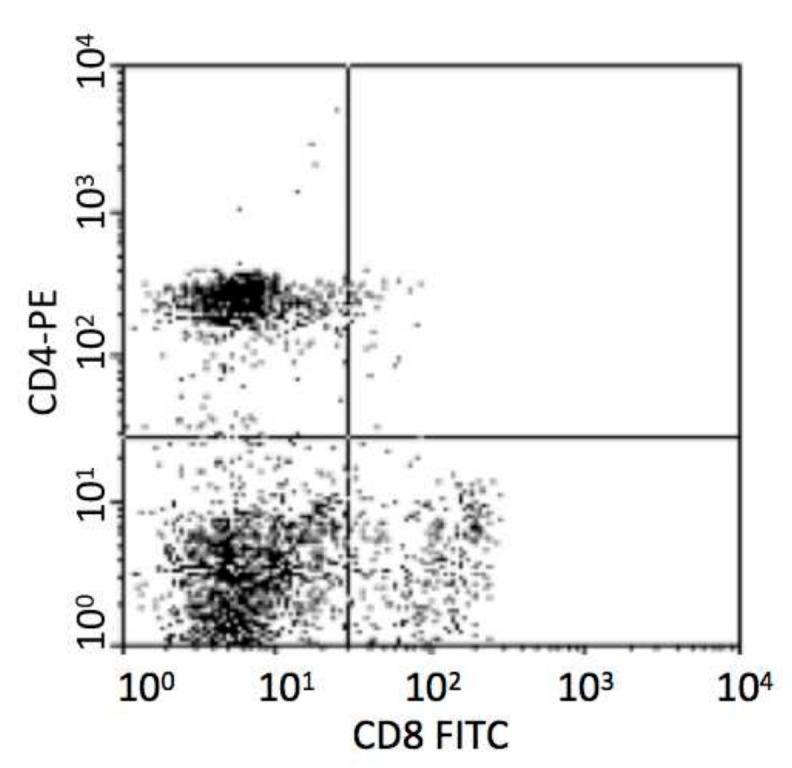
Analysis of porcine CD4+ and CD8+ T cells.

**Figure 2 vaccines-08-00250-f002:**
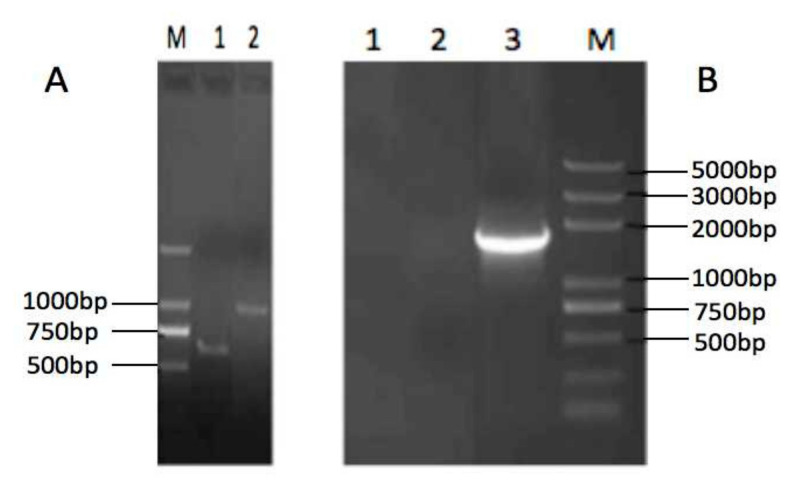
Agarose gel electrophoresis of the Tibetan pig IL-23 p40 and p19 subunits (**A**) (lane M: Trans 2K, lane 1: p19, lane 2: p40) and reverse transcription-polymerase chain reaction (RT-PCR) of VRIL23 transfected cells. (**B**) lane 1 and lane 2: negative control, lane 3: VRIL23.

**Figure 3 vaccines-08-00250-f003:**
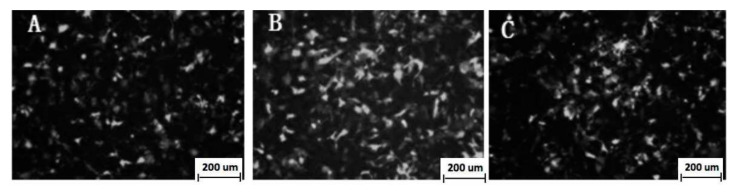
Green fluorescent protein (GFP) fluorescence protein expression in the HEK293 cells (**A**: 24 h; **B**: 48 h; **C**: 72 h).

**Figure 4 vaccines-08-00250-f004:**
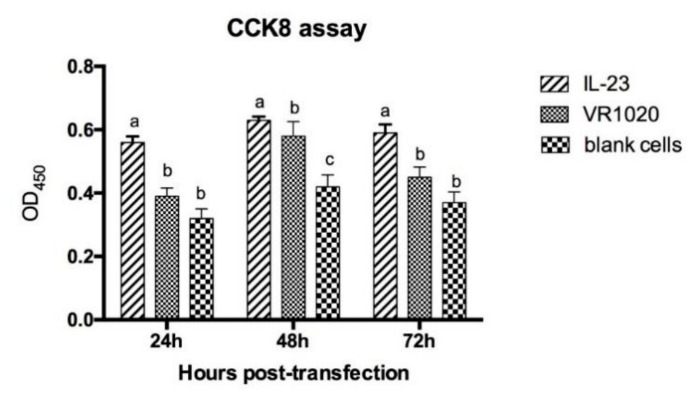
Cell viability assay by CCK8. Different lowercase letters indicate significant difference among different groups at the same time, and vice versa. “a, b and c” means that the data of three groups are significantly different, *p* < 0.05.

**Figure 5 vaccines-08-00250-f005:**
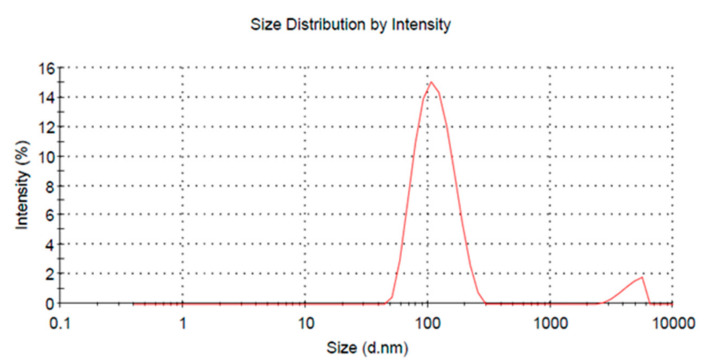
Graph of the VRIL23-chitosan nanoparticles size.

**Figure 6 vaccines-08-00250-f006:**
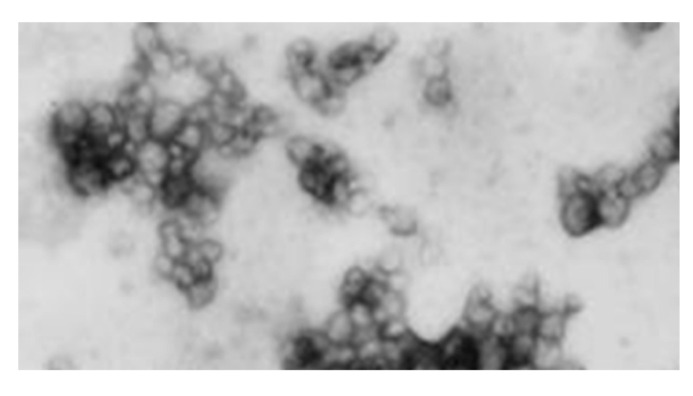
The spherical chitosan nanoparticles under the transmission electron microscope.

**Figure 7 vaccines-08-00250-f007:**
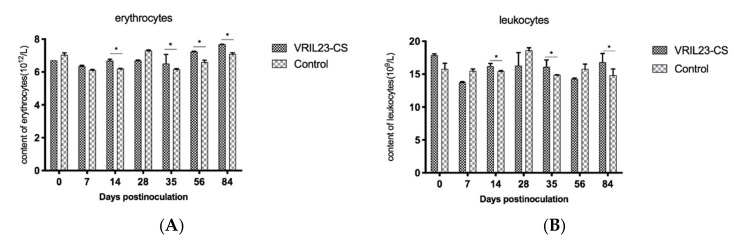
Changes in immune cells from the peripheral blood of the experimental pigs. (**A**) erythrocytes; (**B**)leukocytes; (**C**) hemoglobin; (**D**) platelet. * indicates the difference between the two groups is significant, *p* < 0.05. The followings are the same as here.

**Figure 8 vaccines-08-00250-f008:**
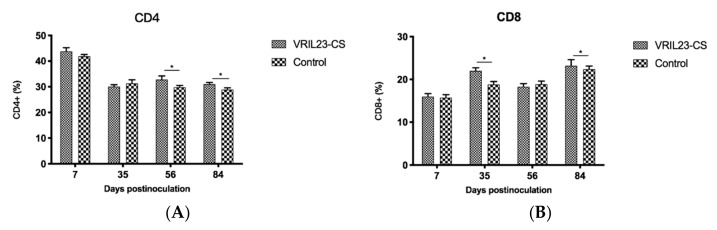
Changes of the CD4^+^ and CD8^+^ T cells in the blood of experimental pigs. (**A**) CD4; (**B**) CD8. * indicates the difference between the two groups is significant, *p* < 0.05.

**Figure 9 vaccines-08-00250-f009:**
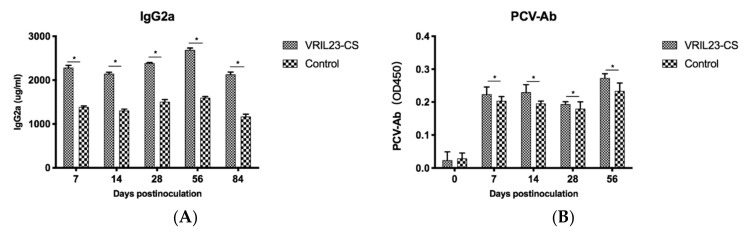
Changes of specific antibody and immunoglobulins in the sera of experimental pigs. (**A**) lgG2a; (**B**) PCV-Ab. * indicates the difference between the two groups is significant, *p* < 0.05.

**Figure 10 vaccines-08-00250-f010:**
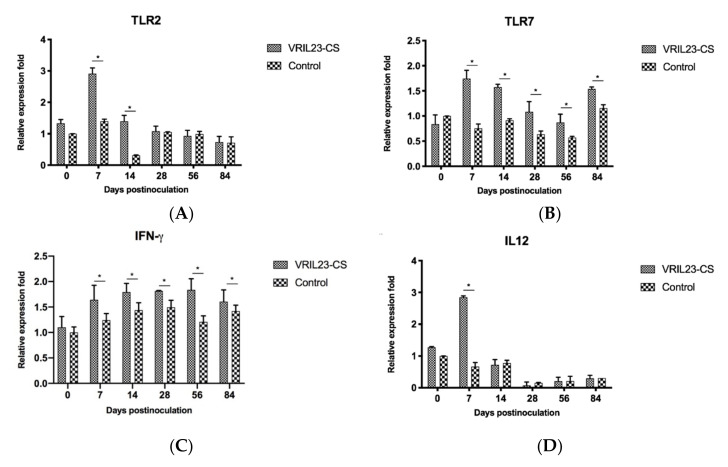
Gene expression levels of TLR2, TLR7, IFN-γ, and IL12 in immune cells of the blood of experimental pigs. (**A**) TLR2; (**B**) TLR7; (**C**) IFN-γ; (**D**) IL12. * indicates the difference between the two groups is significant, *p* < 0.05.

**Figure 11 vaccines-08-00250-f011:**
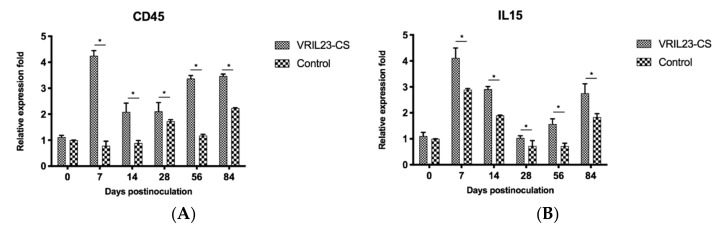
Gene expression levels of CD45 and IL15 in immune cells of the blood of experimental pigs. (**A**) CD45; (**B**) IL15. * indicates the difference between the two groups is significant, *p* < 0.05.

**Figure 12 vaccines-08-00250-f012:**
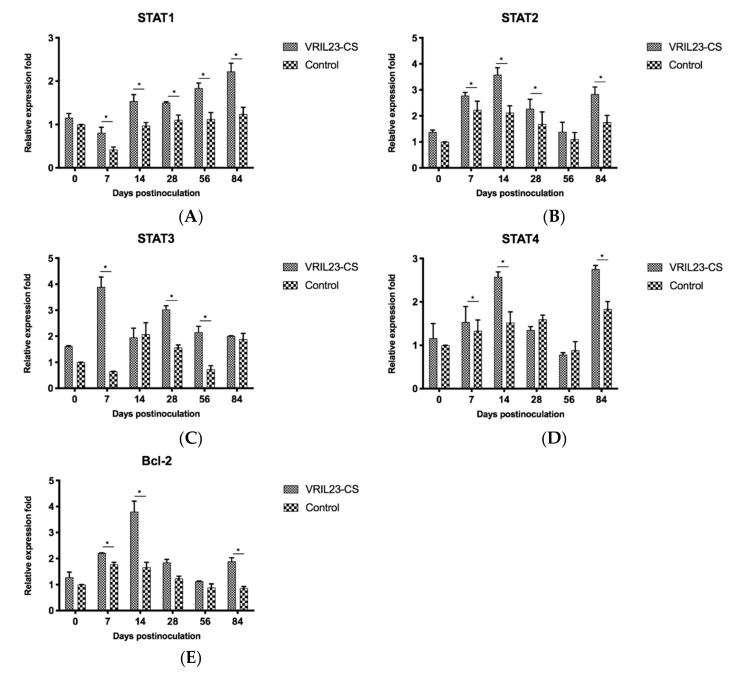
Gene expression levels of immune signal transduction molecules in immune cells of the blood of experimental pigs. (**A**) STAT1; (**B**) STAT2; (**C**) STAT3; (**D**) STAT4; (**E**) Bcl-2. * indicates the difference between the two groups is significant, *p* < 0.05.

**Table 1 vaccines-08-00250-t001:** Primers sequences of p40 and p19.

Name	Sequences (5′-3′)
p40	Forward: ATGCACCTTCAGCAGCTGGTTGTC
	Reverse: CTAATTGCAGGACACAGATGCCCAT
p19	Forward: ATGCTGGGAAGCAGAGCTGTGATG
	Reverse: TTACTGGCTCAGAGTTGCTGCTC

**Table 2 vaccines-08-00250-t002:** Primer sequences and annealing temperatures.

Gene	Primer Sequences (5′-3′)	Annealing Temperature (°C)
PPIA-F	AGACAGCAGAAAACTTCCGTG	52
PPIA-R	ACTTGCCACCAGTGCCATTA	
TLR2-F	TGCTGCAAGGTCAACTCTCT	61
TLR2-R	CAGCAGGGTCACAAGACAGA	
TLR7-F	TTCCTAAAACTCTGCCCTGTG	60
TLR7-R	TTAATGCTGAGGGTGAGGTTG	
IL12-F	ACCCCACATTCCTACTTTTCC	59
IL12-R	TGAGATTTGGTCCGTGAAGAG	
CD45-F	GGACATGTGACCTGGAAACC	55
CD45-R	CCATTACGCTCTGCTTTTCC	
IL15-F	ACTGAGGATGGCATTCATGTC	57.5
IL15-R	GCCAGGTTGCTTCTGTTTTAG	
STAT1-F	TCTGGCACAGTGGCTAGAAAATC	56.3
STAT1-R	GAAAACGGATGGTGGCAAAC	
STAT2-F	AACATTCCTGAGAACCCACTG	54
STAT2-R	CTGTTAGAGACCACGATGAGC	
STAT3-F	AGGACATCAGCGGTAAGA	60
STAT3-R	GGTAGACCAGCGGAGACA	
STAT4-F	CCTGAAAACCCTCTGAAGTACC	53.7
STAT4-R	CTGGGAGCTGTAGTGTTTACC	
Bcl-2-F Bcl-2-R	GAAACCCCTAGTGCCATCAA GGGACGTCAGGTCACTGAAT	60

**Table 3 vaccines-08-00250-t003:** Effect on the weight gain of piglets post inoculation (n = 5).

Group	Initial Weight (kg)	End Weight (kg)	Net Gain (kg)	Average Gain (kg)	Feed Conversion	Death Ratio
VRIL23-CS	7.52 ± 0.36	45.02 ± 2.68	37.50 ± 2.41 ^a^	0.45 ± 0.03 ^a^	2.40	0
Control	7.35 ± 0.25	38.33 ± 4.50	30.98 ± 4.25 ^b^	0.37 ± 0.05 ^b^	2.97	0

Note: Feed conversion = Consuming feed (kg)/Weight gain (kg); the data with different superscript lowercase letters (a and b) in same column imply significant difference (*p* < 0.05).

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
