# Peer review of "Cloning and Expression of the Tibetan Pig Interleukin-23 Gene and Its Promotion of Immunity of Pigs to PCV2 Vaccine"

_vaccines, 2020, doi:10.3390/vaccines8020250_

Round 1
Reviewer 1 Report
General comment
Xiao and coworkers describe cloning and expression of the Tibetan pig interleukin-23 gene and its promotion of immunity of pigs to PCV2 vaccine. The identified differences and results add to our knowledge about the immune response against PCV2. In general, the paper was well written and understandable.
Specific comments/Minor notes
1. Were the animals tested for the presence of PCV3? If they tested, they may want to describe the results
2. Table 3 needs to be supplemented (L.203): should be supplemented with important and useful information, such as feed conversation, deaths ratio, cause of death
Author Response
1.Were the animals tested for the presence of PCV3? If they tested, they may want to describe the results. A: Yes, PCV3 was also tested in animals by ELISA and RT-PCR, along with PCV2, mycoplasma, CSFV and PRRSV, and it was found negative. 2.Table 3 needs to be supplemented (L.203): should be supplemented with important and useful information, such as feed conversation, deaths ratio, cause of death. A: Thanks for your kind suggestion. In results, the section of “Changes of weight” , more information including feed conversation and deaths ratio were added.Reviewer 2 Report
In this research article, authors evaluated adjuvant properties of IL-23 eukaryotic expression vectors associated with chitosan particles (VRIL23-CS) for vaccination against Porcine circovirus type 2 (PCV2). The formulation was inoculated in pig and different parameters were quantified such as antibodies production, immune gene expressions and blood composition. Pigs vaccinated together with VRIL23-CS present higher immune response in comparison with control vaccinated pigs that suggest promising perspectives for PCV2 vaccination. However, some controls are missed in some experiments leading to conclusions that are not supported by the results. This manuscript could be reconsidered for publication in Vaccines (MDPI) after major revision.
Major points:
- Could you provide more information about protective immune response against PCV2 (such as TLRs recognition to justify the selected genes for figures 7 8 9 10 11) and explain deeply why IL-23 expression could be benefit for viral vaccination?
- Authors used anti-CD8 and anti-CD4 to quantify the total number of CD4 T cells and CD8 T cells by flow cytometry. Could you provide the gating strategy and did you exclude dead cells of your analysis? Are you sure that CD8 + and CD4 + cells are only T cells (add CD3 staining)? Finally, could you provide the absolute value of total CD4 and CD8 cells (and not the percentage) (as you quantify the total number of leukocytes, it is possible that the absolute total number of T cells is not higher in VRIL23-CS group to compare with control group depending of the time point) ? As you observed increase of IgG2a antibody production, I wonder if you observe an increase of total blood B cells?
- In all of you kinetic experiments (fig 7 8 9 10 11), I think that a control of non-vaccinated animals or a day 0 is missing. It could be useful to show for example that no PCV-Ab are detected, the basal level of STAT, TLR2 and TLR7 expression…
- As discussed by authors, a control vaccinated group an injected only with chitosan is missed. I consider that this group is essential to support conclusion concerning the potential adjuvant properties of IL-23. The best control group to answer to this question is that one.
Minor points:
- Did you test the endotoxin level of injected VRIL23-CS preparation?
- Line 178: add the dot at the end of the sentence.
- Authors observed upregulation of IL-12 in VRIL23-CS. What about IFN-gamma expression?
- Line 245: replace “adjutants” by “adjuvants”
- Fig 1: figure is blurred and legend need to be aligned
- Fig 2: use green colour to present GFP signal. Add scale on all images. Moreover, could you add a nuclei staining to quantify the percentage of transfected cells?
- Fig 4: how could you explain the second pic? Could you use electronic microscopy to visualize the particles?
Reviewer 3 Report
The authors present the use of a chitosan coated plasmid expressing Tibetan pig Il-23 as adjuvant for PCV2 vaccine. They show that using this plasmid increases the immune response. The manuscript is well written and the results are well presented, with concision.
Control group receives only saline solution and this choice is discussed and justified. However, the use of chitosan empty plasmid for control would have made results more demonstrative.
Expression of Il-23 is not clearly shown. The authors show presence of Il-23 mRNA and expression of GFP but a western showing Il-23 may demonstrate more clearly the expression level of Il-23.
Minor points
- In abstract, line 26 “increased” would be better than “mounted”
- Line 78 “linkage of p40 and p19 was finished…”: “achieved” would be better than “finished”
- Line 96 and 97: “hole” should be replaced by “well”
- Line 270 “chitosan”
Author Response
- In abstract, line 26 “increased” would be better than “mounted”
A: Yes, the statement of “increased” were corrected as “mounted”.
- Line 78 “linkage of p40 and p19 was finished…”: “achieved” would be better than “finished”
A: Yes, “finished” was replaced by “achieved”.
- Line 96 and 97: “hole” should be replaced by “well”
A: Yes, the statement of “hole” was corrected as “well”.
- Line 270 “chitosan”
A: Yes, chitosan was used.
Round 2
Reviewer 2 Report
Authors addressed all my concerns. The manuscript is suitable for publication. However, I recommend adding the two figures presented in the “response to reviewer 2” into the revised manuscript.
Author Response
Dear reviewer:
Thanks for your kind suggestion.
I have already added the two figures presented in the first round of “response to reviewer 2” into this revised manuscript.
Please see the attachment.